# Structural basis of *trans*-synaptic interactions between PTPδ and SALMs for inducing synapse formation

Sakurako Goto-Ito[1,2,3], Atsushi Yamagata[1,2,3,4], Yusuke Sato[1,2,3,4], Takeshi Uemura[3,5,6], Tomoko Shiroshima[1,2,3], Asami Maeda[1,2,3], Ayako Imai[7], Hisashi Mori [7], Tomoyuki Yoshida[7,8] & Shuya Fukai [1,2,3,4]

Synapse formation is triggered by *trans*-synaptic interactions of cell adhesion molecules, termed synaptic organizers. Three members of type-II receptor protein tyrosine phosphatases (classified as type-IIa RPTPs; PTPδ, PTPσ and LAR) are known as presynaptic organizers. Synaptic adhesion-like molecules (SALMs) have recently emerged as a family of postsynaptic organizers. Although all five SALM isoforms can bind to the type-IIa RPTPs, only SALM3 and SALM5 reportedly have synaptogenic activities depending on their binding. Here, we report the crystal structures of apo-SALM5, and PTPδ–SALM2 and PTPδ–SALM5 complexes. The leucine-rich repeat (LRR) domains of SALMs interact with the second immunoglobulin-like (Ig) domain of PTPδ, whereas the Ig domains of SALMs interact with both the second and third Ig domains of PTPδ. Unexpectedly, the structures exhibit the LRR-mediated 2:2 complex. Our synaptogenic co-culture assay using site-directed SALM5 mutants demonstrates that presynaptic differentiation induced by PTPδ–SALM5 requires the dimeric property of SALM5.

[1] Institute of Molecular and Cellular Biosciences, The University of Tokyo, Tokyo 113-0032, Japan. [2] Synchrotron Radiation Research Organization, The University of Tokyo, Tokyo 113-0032, Japan. [3] CREST, JST, Saitama 332-0012, Japan. [4] Department of Medical Genome Sciences, Graduate School of Frontier Sciences, The University of Tokyo, Chiba 277-8501, Japan. [5] Department of Molecular and Cellular Physiology, Shinshu University School of Medicine, Nagano 390-8621, Japan. [6] Department of Biological Sciences for Intractable Neurological Diseases, Institute for Biomedical Sciences, Interdisciplinary Cluster for Cutting Edge Research, Shinshu University, Nagano 390-8621, Japan. [7] Department of Molecular Neuroscience, Graduate School of Medicine and Pharmaceutical Sciences, University of Toyama, Toyama 930-0194, Japan. [8] PRESTO, JST, Saitama 332-0012, Japan. Correspondence and requests for materials should be addressed to T.Y. (email: toyoshid@med.u-toyama.ac.jp) or to S.F. (email: fukai@iam.u-tokyo.ac.jp)

The mammalian brain contains at least 100 billion neurons, which are connected via synapses to form comprehensive networks for brain functions. The differentiation of synapses in neuronal development can be induced by receptor-like adhesion molecules, termed synaptic organizers. Dysfunctions of synaptic organizers potentially cause neurodevelopmental disorders such as autism spectrum disorders (ASD), intellectual disability, or schizophrenia[1–4]. Trans-synaptic interactions between pre- and postsynaptic organizers can induce synapse formation[5,6]. Type-IIa receptor protein tyrosine phosphatases (RPTPs) and neurexins are the two major presynaptic organizers[3–5,7]. In mammals, the type-IIa RPTPs have three members, PTPδ, PTPσ, and LAR. These members possess a large extra-cellular domain (ECD) consisting of three immunoglobulin-like (Ig) domains and four to eight fibronectin type-III (Fn) domains, followed by a single transmembrane helix and a cytoplasmic domain harboring two protein tyrosine phosphatase (PTP) domains (Fig. 1a): the first PTP domain is active, while the second domain is inactive. The ECDs of the type-IIa RPTPs bind to those of various postsynaptic organizers such as the interleukin-1 receptor accessory protein (IL-1RAcP)[8], IL-1RAcP-like-1 (IL1RAPL1)[9], netrin-G ligand-3 (NGL-3)[10,11], neurotrophin receptor tyrosine kinase C (TrkC)[12], and Slit- and Trk-like family proteins (Slitrks)[13,14]. The binding of the type-IIa RPTPs with IL-1RAcP, IL1RAPL1, and Slitrks is controlled by the two short peptide inserts derived from alternative splicing of the type-IIa RPTP genes at the sites corresponding to a loop within Ig2 (mini-exon A; meA) and the junction between Ig2 and Ig3 (mini-exon B; meB; Fig. 1a). In PTPδ that is expressed in the brain, there are four variations of the meA peptide (ESIGGTPIR (A9), GGTPIR, ESI or none (−)) and two variations of the meB peptide (ELRE (+) or none (−)). PTPσ has two variations in the meB peptide, whereas LAR has two variations in both the meA and meB peptides[9]. The complex structures between PTPδ and IL-1RAcP, IL1RAPL1 or Slitrks have provided the structural and mechanistic insights into their splicing-dependent trans-synaptic interactions for inducing synapse formation[9,15–17].

Synaptic adhesion-like molecules (SALMs; also known as Lrfns) family proteins have recently emerged as a family of postsynaptic organizers that bind to the type-IIa RPTPs[18–20]. SALM family proteins have five isoforms (SALM1–SALM5). Their ECDs share the same domain organization consisting of a leucine-rich repeat (LRR) domain, an Ig domain and an Fn domain with 35% amino-acid sequence identity (Fig. 1a). In contrast, their intracellular regions are diverged (<3% sequence identity). SALM1–SALM3, but not SALM4 or SALM5, possess PSD-95/DLG1/ZO-1 (PDZ)-binding motifs. An essential post-synaptic scaffold protein, PSD-95, can bind to the PDZ-binding motifs of SALM1–SALM3 in vitro and form a complex with those of SALM1 and SALM2 in vivo[21–23]. The SALM1 complex includes PSD-95 and the GluN1 subunit of NMDA receptor[23], whereas SALM2 immunoprecipitates with PSD-95 and the GluA1/A2 subunits of AMPA receptors[21]. SALM1-null mice display morphologically abnormal synapses in the hippocampus and show ASD-like behaviors with enhanced cognitive function[22].

Previous studies have shown the differences in the functional roles of SALM1–SALM5 in neuronal development. Although all isoforms can bind to the type-IIa RPTPs, only SALM3 and SALM5 are capable of inducing excitatory and inhibitory pre-synaptic differentiation in contacting axons[19,24]. The number of excitatory synapses in the hippocampal CA1 region is obviously reduced in SALM3-null mice[18]. Knockdown of SALM5 reduces the number of excitatory and inhibitory synapses[24]. SALM5 has been reported to be associated with ASD, intellectual disability, and schizophrenia in humans[25–28].

Despite the functional importance of trans-synaptic adhesions mediated by SALMs and the type-IIa RPTPs in neuronal development, their underlying structural mechanisms remain elusive. In this study, we present the crystal structures of apo-SALM5 and the PTPδ–SALM2 and PTPδ–SALM5 complexes. Together with structure-based mutagenesis, in vitro binding analysis with surface-plasmon resonance (SPR) spectroscopy and synaptogenic co-culture assay, we reveal the structural basis of trans-synaptic interactions between the SALMs and type-IIa RPTPs.

## Results

**Structures of PTPδ–SALM2 and PTPδ–SALM5 complexes**. We determined the crystal structures of apo-SALM5 (LRR–Ig), and PTPδ (Ig1–Ig3)–SALM2 (LRR–Ig) and PTPδ (Ig1–Fn1)–SALM5 (LRR–Ig) complexes at 3.08, 3.16, and 4.18 Å resolutions, respectively (Fig. 1 and Table 1). For crystallization, a mouse PTPδ isoform containing both meA9 and meB inserts, and human SALM5 and mouse SALM2 were used because the expression level of human SALM5 LRR–Ig was much higher than that of mouse SALM5 LRR–Ig. In both the PTPδ–SALM2 and PTPδ–SALM5 crystals, there are two complexes in the asymmetric unit (Fig. 1b). The structures of the two complexes in each crystal were nearly identical (rmsds of 0.94 and 1.1 Å for PTPδ–SALM2 and PTPδ–SALM5, respectively).

The LRR–Ig structures of SALM2 and SALM5 in their complexes with PTPδ are similar with rmsds of 1.41–1.70 Å over a span of 285–330 residues (Fig. 1c). In the apo-SALM5 structure, the electron density of the Ig domain was mostly invisible, probably owing to the structural disorder. The electron densities of the Ig domains of SALM2 and SALM5 became relatively clear upon binding to PTPδ, suggesting that the Ig domains of SALM2 and SALM5 were partly stabilized by the interaction with PTPδ. The LRR domains of SALM2 and SALM5 are composed of eight parallel β-strands flanked by the N- and C-terminal caps. Ten residues between the seventh and eighth repeats were disordered. The Ig domains of SALM2 and SALM5 are positioned on the convex surface of the LRR domain and stabilized by a disulfide bond tethering the C-terminal cap and the linker connecting the LRR and Ig domains.

The structure of each domain of PTPδ in the PTPδ–SALM2 and PTPδ–SALM5 complexes is basically identical to those in other synaptic organizer complexes with PTPδ[15–17,29]. Although the Ig1 and Ig2 domains of PTPδ form a similar V-shaped unit, the relative positions and orientations of Ig3 and Fn1 are different from those in other PTPδ or PTPσ complexes[15–17,29,30] (Supplementary Fig. 1). In the PTPδ–SALM5 complex, the electron density of PTPδ Fn1 in one complex in the asymmetric unit was unclear, probably owing to the structural disorder. No electron density of the meA9 insert ([188]ESIGGTPIR[196]) in PTPδ Ig2 was observed in either the PTPδ–SALM2 or PTPδ–SALM5 complex, suggesting that meA is completely disordered and not involved in the binding of PTPδ to SALM2 or SALM5. Consistently, our SPR analysis showed that the meA-containing and meA-lacking PTPδ isoforms (A9B+ and A−B+, respectively) bind to SALM5 with similar affinities ($K_D$; Table 2 and Supplementary Fig. 2). The meB insert ([234]ELRE[237]) resides in the junction between Ig2 and Ig3 of PTPδ (Fig. 1a, b). Although meB has no obvious interaction with SALM5, it is located close to SALM2 and SALM5. The contribution of meB to the binding to SALM2 and SALM5 is discussed in the next section.

**Interactions of PTPδ with SALM2 and SALM5**. The binding modes of the PTPδ and SALM ECDs are quite similar between the PTPδ–SALM2 and PTPδ–SALM5 complexes (Fig. 1b): the

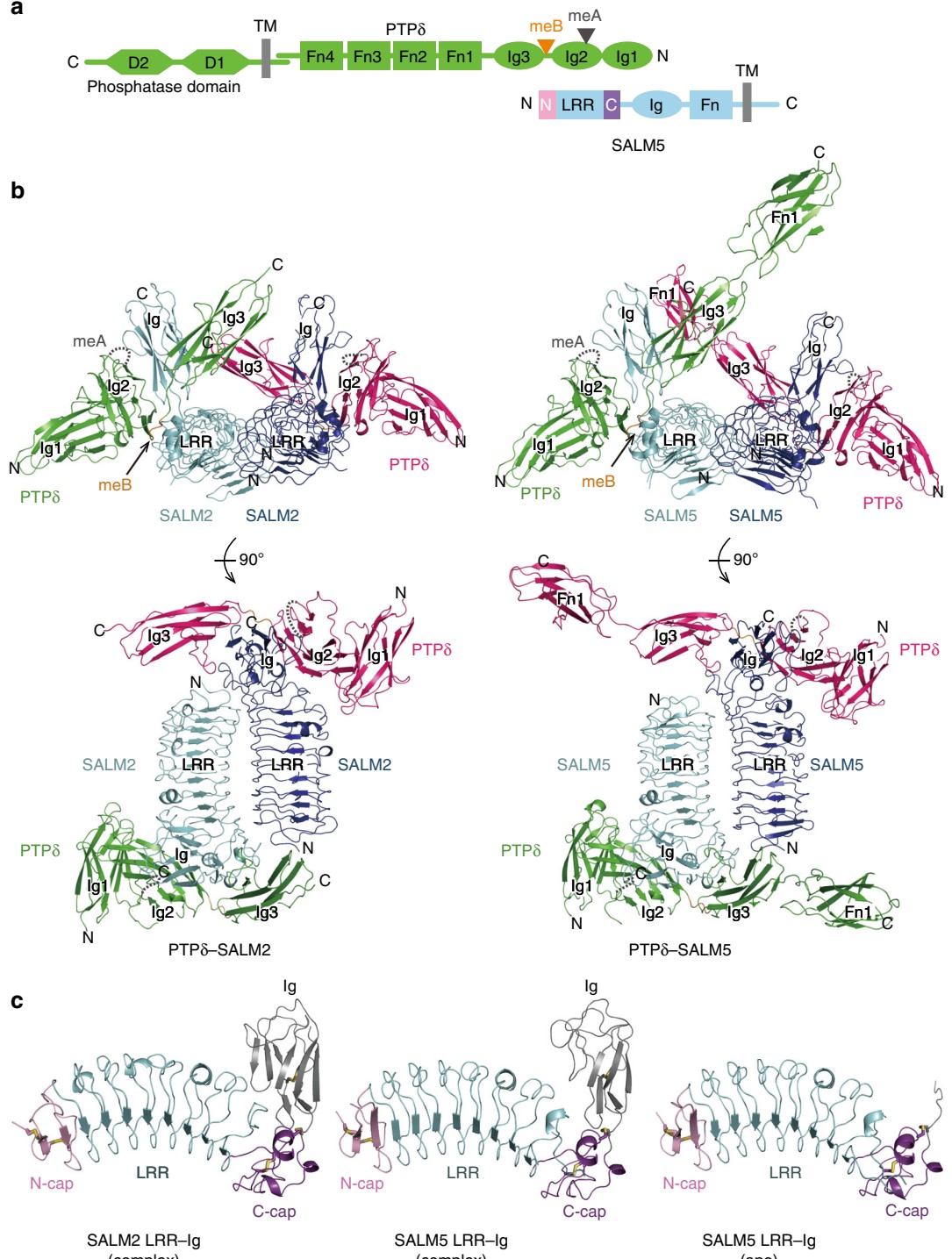

**Fig. 1** Structures of PTPδ–SALM2 and PTPδ–SALM5 complexes. **a** Domain organizations of PTPδ and SALM5. **b** Structures of PTPδ–SALM2 and PTPδ–SALM5 complexes. Two complexes in the asymmetric unit are shown in two different orientations. PTPδ and SALMs in one complex are colored in green and cyan, respectively, whereas those in the other complex are colored in magenta and deep blue, respectively. The meB insertion in PTPδ are colored in orange. The disordered meA insertion is drawn as grey dotted lines. **c** LRR-Ig structures of SALM2 and SALM5. N-cap, LRR, C-cap, and Ig are colored in pink, cyan, purple, and grey, respectively. The disulfide bonds are shown as yellow sticks

Ig2 and Ig3 domains of PTPδ sandwiches the Ig domain of SALM2 or SALM5. The Ig2 domain of PTPδ also interacts with the LRR domain of SALM2 or SALM5 (Fig. 2). The PTPδ-interacting residues are conserved or functionally equivalent between SALM2 and SALM5 (Supplementary Table 1 and Supplementary Fig. 3). On the other hand, only SALM5 can induce presynaptic differentiation by binding to the type-IIa RPTPs[24]. Assessment of the structure–function relationship by structure-guided site-directed mutagenesis and synaptogenic co-culture assay could be applied to SALM5 but not to SALM2. Therefore, we hereafter describe the interaction in the SALM5–PTPδ complex, unless otherwise noted.

**Table 1 Data collection and refinement statistics**

| Molecule name | SALM5 (LRR–Ig) | PTPδ (Ig1–Fn1)–SALM5 (LRR–Ig) | PTPδ (Ig1–Ig3)–SALM2 (LRR–Ig) |
|---|---|---|---|
| PDB ID | 5XWS | 5XWT | 5XWU |
| *Data collection* | | | |
| Beamline | PF BL-1A | SPring-8 BL41XU | SPring-8 BL41XU |
| Space group | $P6_322$ | $P2_12_12_1$ | $P2_12_12_1$ |
| Cell constants | | | |
| *a, b, c* (Å) | 154.9, 154.9, 91.3 | 98.3, 169.8, 210.9 | 90.0, 127.2, 210.9 |
| *α, β, γ* (°) | 90, 90, 120 | 90, 90, 90 | 90, 90, 90 |
| Resolution | 50–3.08 (3.15–3.08) | 50–4.18 (4.27–4.18) | 50–3.16 (3.23–3.16) |
| $R_{sym}$ | 0.113 (1.18) | 0.220 (0.565) | 0.198 (0.597) |
| $I/\sigma I$ | 20.0 (1.77) | 6.24 (1.94) | 7.48 (1.59) |
| Redundancy | 16.9 (16.3) | 7.1 (5.4) | 9.6 (7.0) |
| Completeness (%) | 100 (100) | 96.9 (93.2) | 99.8 (99.7) |
| *Refinement* | | | |
| Resolution | 45–3.08 | 50–4.18 | 49–3.16 |
| No. reflections | 12,253 | 25,606 | 41,611 |
| $R_{work}/R_{free}$ | 0.268/0.308 | 0.261/0.311 | 0.218/0.259 |
| No. atoms | | | |
| Protein | 2,153 | 11,289 | 9,809 |
| Sugar | 28 | 238 | 140 |
| MES | — | — | 24 |
| *B*-factors (Å²) | | | |
| Protein | 73.4 | 136.9 | 56.0 |
| Sugar | 150.5 | 183.8 | 112.1 |
| MES | — | — | 70.7 |
| Rmsds | | | |
| Bond lengths (Å) | 0.002 | 0.006 | 0.005 |
| Bond angles (°) | 0.74 | 0.66 | 0.72 |
| Ramachandran plot (%) | | | |
| Favored | 93.6 | 91.8 | 93.2 |
| Allowed | 6.4 | 8.2 | 6.8 |
| Outliers | 0.0 | 0.0 | 0.0 |

Values in parentheses are for the highest resolution shell

**Table 2 Binding affinities between PTPδ splicing variants (Ig1–Fn1) and SALM5 (LRR–Ig)**

| PTPδ | $K_D$ (μM) | Relative affinity (%) |
|---|---|---|
| A9B+ | 14.4 ± 3.2 | 100 |
| A–B+ | 13.5 ± 0.8 | 107 |
| A9B– | 105.8 ± 17.0 | 13.6 |

The interactions between PTPδ and SALM5 occur at three interfaces (Fig. 2a). The most extensive interface is formed between PTPδ Ig2 and SALM5 LRR (Fig. 2a, b). The side chain of Gln209 and the main chain of Val232 in PTPδ hydrogen bond with Arg253 of SALM5. Arg233 of PTPδ hydrogen bonds with the main chains of Glu279, Glu280, and Phe282 in SALM5. In addition to these hydrogen bonds, hydrophobic interactions are formed by Leu141, Val143, and Tyr231 of PTPδ, and Leu249 and Trp250 of SALM5. The contributions of these interacting residues to the binding between PTPδ and SALM5 were assessed by SPR analyses of their site-directed mutants (Table 3 and Supplementary Fig. 4). The R253A mutation of SALM5 decreased the affinity to 15% of wild-type affinity. On the other hand, the Q209A mutant of PTPδ retained 66% of wild-type affinity, probably because the hydrogen bond with the main chain of PTPδ Val232 still remained. The R233A mutant of PTPδ retained 58% of wild-type affinity, indicating that the hydrogen bonds mediated by Arg253 of PTPδ contribute little to the binding to SALM5. As compared with the hydrogen bonds, hydrophobic interactions are more critical for the binding: the L141A mutation of PTPδ

decreased the affinity to an unmeasurable level. The V143A and Y231A mutations of PTPδ decreased the affinities to 9.6 and 20% of wild-type affinity, respectively, whereas the L249A mutation of SALM5 decreased the affinity to 8.5% of wild-type affinity. The contribution of SALM5 Trp250 to the binding could not be evaluated, because the W250A mutant of SALM5 was defective in expression and/or secretion.

The second interface is formed between PTPδ Ig2 and SALM5 Ig (Fig. 2a, c). A hydrogen bond is formed between the main chains of PTPδ Val143 and SALM5 Pro362. Although Pro362 of SALM5 is replaced by Ala375 in SALM2, a similar hydrogen bond is formed between the main chains of PTPδ Val143 and SALM2 Ala375 in the PTPδ–SALM2 complex (Supplementary Fig. 3b). In addition to this hydrogen bond, hydrophobic interactions are formed by Met137 and Leu153 of PTPδ, and Ile321 and Ile358 of SALM5. The single Ala replacements of these residues decreased the affinities to 36–65% of wild-type affinity (Table 3 and Supplementary Fig. 4). Therefore, the hydrophobic interactions at the PTPδ Ig2–SALM5 Ig interface appear to be less important for the binding between PTPδ and SALM5 than those at the PTPδ Ig2–SALM5 LRR interface.

The third interface is formed between PTPδ Ig3 and SALM5 Ig (Fig. 2a, d). This interface is primarily hydrophobic: Tyr273 and Met312 of PTPδ form a hydrophobic patch, which faces Leu288 of SALM5. The Y273A and M312A mutations of PTPδ and the L288A mutation of SALM5 decreased the affinities to 19%, 39%, and 11% of wild-type affinity, respectively (Table 3 and Supplementary Fig. 4). The hydrophobic interaction between PTPδ Tyr273 and SALM5 Leu288 contributes to the binding

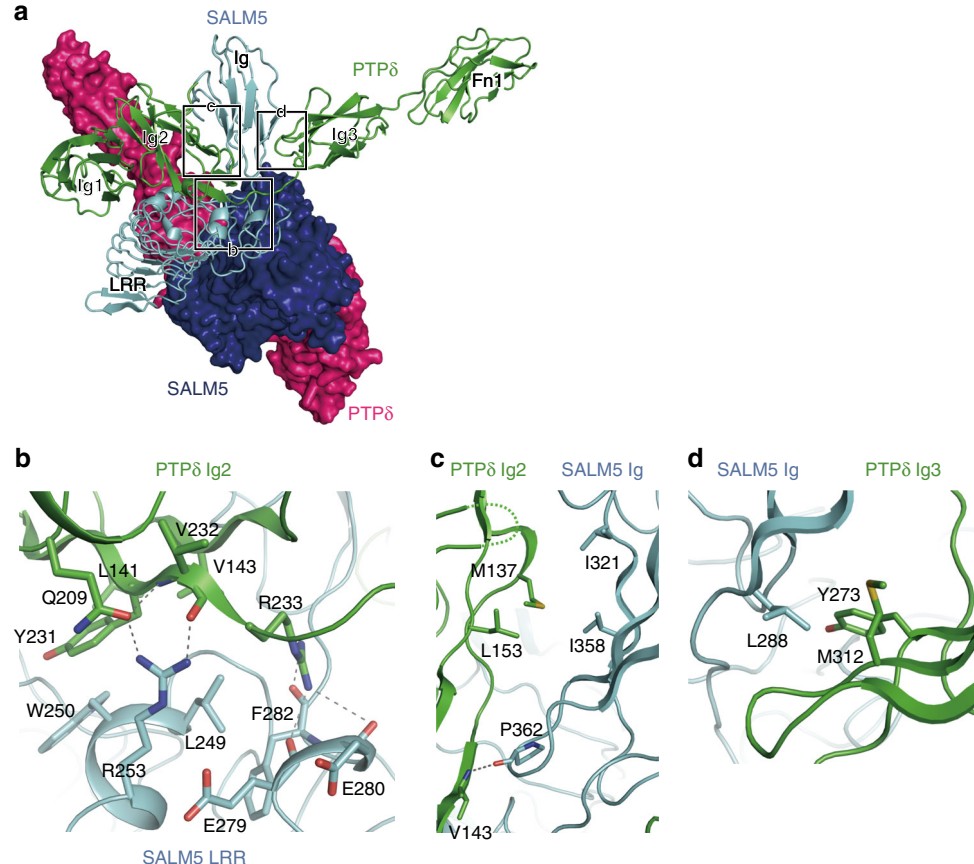

**Fig. 2** Interfaces between PTPδ and SALM5. Hydrogen bonds are indicated as dotted lines. **a** Overall view of 2:2 PTPδ–SALM5 complex. One PTPδ–SALM5 complex is shown as a cartoon model with PTPδ and SALM5 colored in green and cyan, respectively, whereas the other complex is shown as a surface model with PTPδ and SALM5 colored in magenta and deep blue, respectively. The interfaces between PTPδ and SALM5 in the cartoon model are enclosed by three rectangles. The details of each interface are shown in **b–d**, as indicated. **b** Interface between PTPδ Ig2 and SALM5 LRR. **c** Interface between PTPδ Ig2 and SALM5 Ig. **d** Interface between PTPδ Ig3 and SALM5 Ig

| Table 3 Binding affinities between PTPδ (Ig1-Fn1) and SALM5 (LRR-Ig) | | | |
|---|---|---|---|
| **PTPδ** | **SALM5** | $K_D$ (μM) | **Relative affinity (%)** |
| WT | WT | 14.4 ± 3.2 | 100 |
| *PTPδ Ig2–SALM5 LRR* | | | |
| Q209A | WT | 21.9 ± 0.23 | 66 |
| R233A | WT | 24.8 ± 1.2 | 58 |
| WT | R253A | 95.2 ± 1.7 | 15 |
| L141A | WT | N.D. | — |
| V143A | WT | 150.3 ± 49.5 | 9.6 |
| Y231A | WT | 70.9 ± 7.8 | 20 |
| WT | L249A | 169.1 ± 7.5 | 8.5 |
| *PTPδ Ig2–SALM5 Ig* | | | |
| M137A | WT | 39.9 ± 3.4 | 36 |
| L153A | WT | 25.0 ± 0.27 | 58 |
| WT | I321A | 21.9 ± 2.1 | 66 |
| WT | I358A | 31.4 ± 5.4 | 46 |
| *PTPδ Ig3–SALM5 Ig* | | | |
| Y273A | WT | 77.5 ± 0.63 | 19 |
| M312A | WT | 37.3 ± 1.1 | 39 |
| WT | L288A | 129.4 ± 2.2 | 11 |
| WT wild type, N.D. not detectable | | | |

between PTPδ and SALM5 as much as those at the PTPδ Ig2–SALM5 LRR interface. The meB-containing linker connecting Ig2 and Ig3 of PTPδ is stretched along SALM5, enabling PTPδ Ig3 to interact with SALM5 Ig (Fig. 1b). We therefore determined the affinity of the meB-lacking PTPδ isoform (A9B−) to SALM5 LRR–Ig. A9B− had a much lower (14%) affinity than A9B+ (Table 2 and Supplementary Fig. 2). The meB insertion likely plays an important role in placing PTPδ Ig3 at the position favorable for the interaction with SALM5 Ig.

**LRR-mediated dimer of SALM2 and SALM5.** The LRR domains of SALM2 and SALM5 in the asymmetric units of the crystals are aligned in an antiparallel fashion with their long sides facing each other (Figs. 1b and 3a). These dimer-like LRR–LRR interactions of SALM2 and SALM5 are highly similar. Furthermore, in the apo-SALM5 crystal, a similar dimer-like interaction was also observed between two adjacent molecules related by crystallographic symmetry (Fig. 3a and Supplementary Fig. 5a). We therefore tested the possibility of homodimer formation of SALM2 and SALM5 LRR–Ig molecules in solution by size-exclusion chromatography coupled with multi-angle laser light scattering (SEC-MALS). The theoretical molar masses of the monomeric SALM2 and SALM5 LRR–Ig-His$_6$ molecules are 40.2 and 41.2 kDa, whereas the molar masses of the SALM2 and SALM5 LRR–Ig-His$_6$ molecules determined by SEC-MALS are 77.0 and 77.1 kDa, respectively, indicating that both SALM2 and SALM5 form homodimers in solution (Fig. 3d).

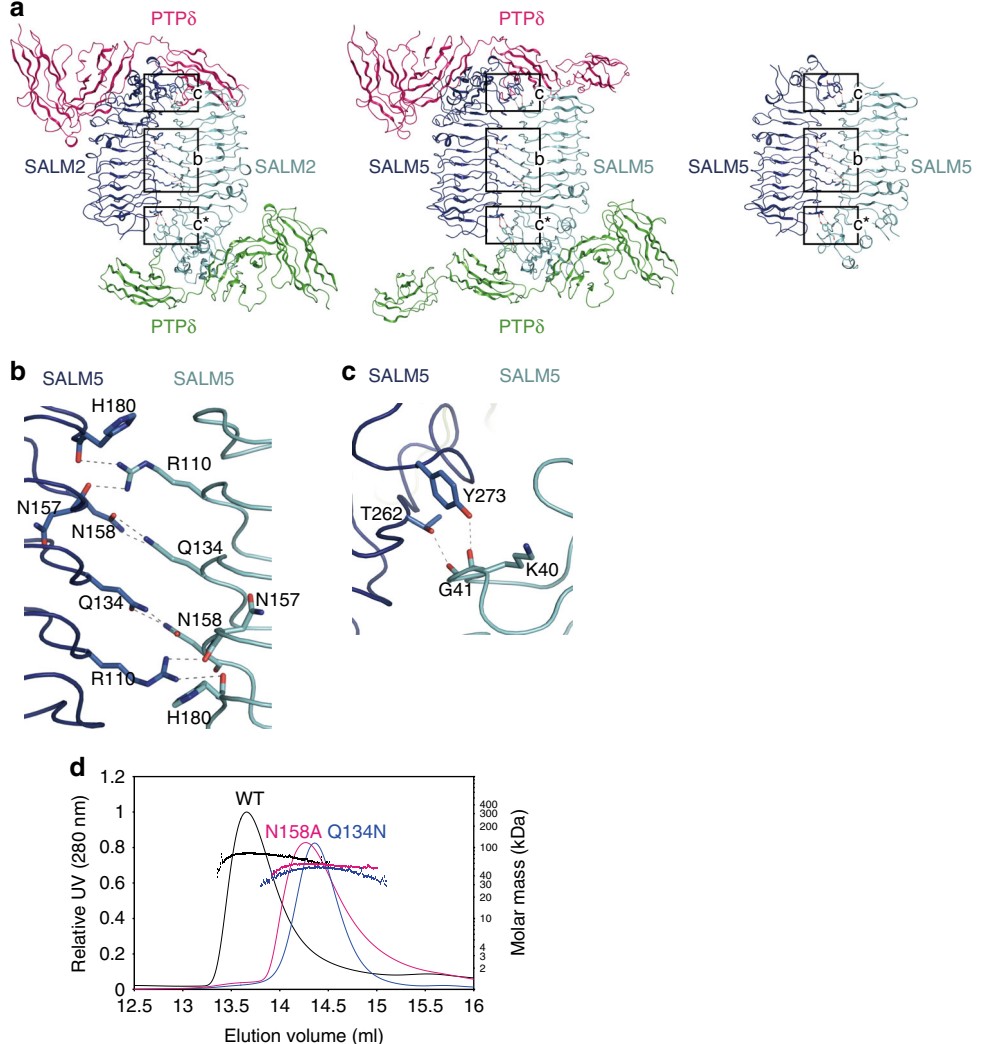

**Fig. 3** LRR-mediated dimer of SALM5. **a** Overall views of the 2:2 complexes of PTPδ–SALM2 and PTPδ–SALM5 and the dimer of apo SALM5. The central and peripheral regions of the dimer interface are enclosed by the rectangles labeled 'b' and 'c', respectively. The details are shown in **b** and **c**, as indicated. The regions 'c' and 'c*' are related by pseudo-*C*2 (PTPδ–SALM2 and PTPδ–SALM5) or *C*2 (apo-SALM5) symmetry. **b** Interactions in the central region of the SALM5 dimer interface. Hydrogen bonds are indicated as dotted lines. **c** Interactions in the peripheral region of the SALM5 dimer interface. Hydrogen bonds are indicated as dotted lines. **d** SEC-MALS analyses of wild type and N158A and Q134N mutants of SALM5 LRR–Ig (black, magenta, and blue, respectively). The plotted dots that look like dotted lines correspond to the calculated molar masses of each protein

The residues involved in the LRR–LRR dimer interface are conserved between SALM2 and SALM5 (Supplementary Table 1, Fig. 3b, c and Supplementary Fig. 5b, c). We hereafter describe the dimer interface in the SALM5–PTPδ complex, unless otherwise noted. A pseudo-*C*2 symmetric hydrogen bond network is extensively formed in the dimer interface. In the central region of the interface, Arg110 of one molecule hydrogen bonds with the main chains of His180 and Asn157 in another molecule (hereafter denoted by * after residue numbers), whereas Asn158 hydrogen bonds with Gln134* (Fig. 3b). In the peripheral region of the interface, the main chains of Lys40 and Gly41 hydrogen bond with Thr262* and Tyr273*, respectively (Fig. 3c).

To evaluate the importance of the residues involved in hydrogen bonding in the central dimer interface, the N158A and Q134N mutants of SALM5 were analyzed by SEC-MALS (Fig. 3d). We could not analyze the Q134A or R110A/S/N/Q mutant of SALM5 owing to their severe aggregations. The majorities of the N158A and Q134N mutants behaved as monomers with the determined molar masses of 55.8 and 49.1

kDa, respectively. This result indicates that the hydrogen bonds in the central region of the dimer interface are critical for the dimer formation of SALM5 and also confirms the dimerization property of SALM5 in solution.

**Synaptogenic activity of SALM5.** Since SALM5 is thought to be a unidirectional synapse organizer that induces presynaptic differentiation[24], we next examined the effects of these mutations of SALM5 on the presynapse-inducing activity using co-cultures of cerebral cortical neurons and magnetic beads conjugated with the SALM5 mutants fused to Fc. The induction of the presynaptic differentiation of cortical neurons contacting the beads was evaluated by immunostaining of the presynaptic active zone protein Bassoon (Fig. 4). Compared with the marked accumulation of strong Bassoon signals on the beads coated with wild-type SALM5-Fc, the mutations in the SALM5 LRR on the interface of PTPδ Ig2 (L249A and R253A) abolished the accumulation of Bassoon around the beads. Furthermore, the mutations in the SALM5 Ig on the interface of both PTPδ Ig2 (I358A) and PTPδ Ig3 (L288A)

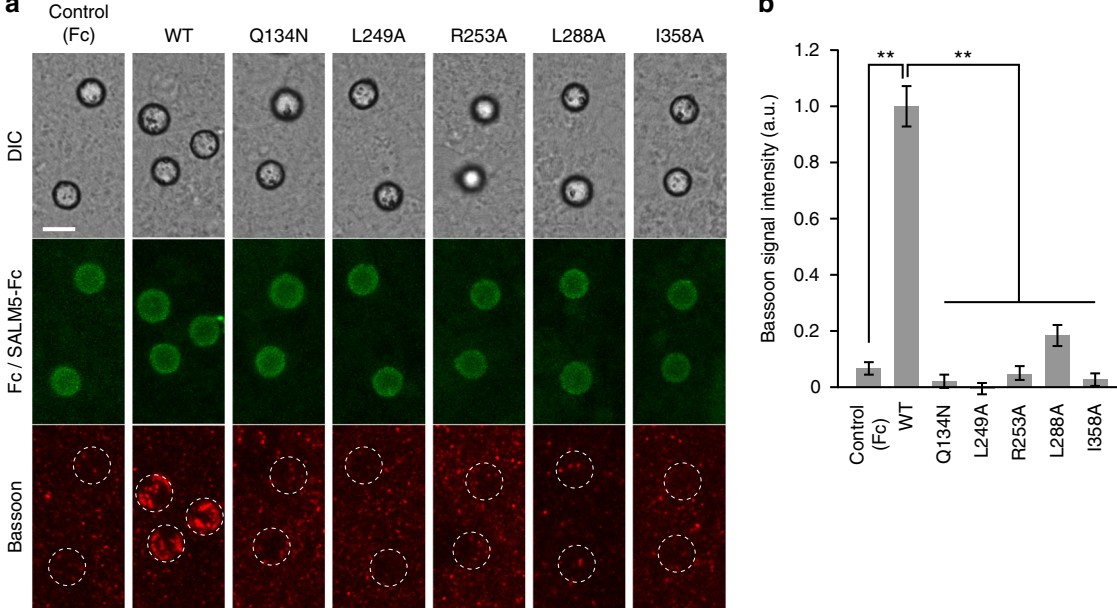

**Fig. 4** Presynapse-inducing activities of SALM5 mutants. **a** Co-cultures of cortical neurons and beads conjugated with Fc or SALM5 mutants fused to Fc. Co-cultures were stained with anti-Fc (middle row, green) and anti-Bassoon (bottom row, red) antibodies. Corresponding differential interference contrast (DIC) images are on the top row. Scale bar, 5 μm. **b** Staining signal intensities of Bassoon on beads conjugated with Fc or SALM5 mutants fused to Fc. All values represent mean ± s.e.m. Statistical significance was evaluated by one-way ANOVA followed by post hoc Tukey's test. **$p < 0.01$, compared with wild-type SALM5-Fc. $n = 28$–33 beads

suppressed the synaptogenic activity. The suppressive effect of these SALM5 interface mutations on synaptogenic activity correlated well with the binding affinities measured by SPR (Fig. 4b, Table 3 and Supplementary Fig. 4). In contrast, the Q134N mutation, which interferes with the dimerization of SALM5 but hardly affects the binding to PTPδ ($K_D = 9.0 \pm 1.6$ μM; Supplementary Fig. 4), severely reduced the synaptogenic activity, suggesting that SALM5-mediated dimerization occurs under physiological conditions and may be a prerequisite for the type-IIa RPTPs to elicit signals for presynaptic differentiation.

## Discussion

The PTPδ residues involved in the binding to SALM2 and SALM5 are completely conserved among the type-IIa RPTPs (Supplementary Fig. 6). Similarly, the PTPδ-interacting residues of SALM2 and SALM5 are conserved or replaced by functionally equivalent residues among SALMs (Supplementary Fig. 7). These sequence conservations suggest that PTPδ, PTPσ and LAR bind to SALM1–SALM5 in the same manner. Consistently, the mutations of the PTPδ-interacting residues of SALM5 completely suppressed the synaptogenic activity to cortical neurons that express all the type-IIa RPTPs (Fig. 4). In addition, the residues involved in the dimer interactions of SALM2 and SALM5 are completely conserved among SALMs (Supplementary Fig. 7). This sequence conservation suggests that SALMs form not only a homodimer but also a heterodimer consisting of two different SALM isoforms, as proposed previously[31,32]. The various combinations of SALM dimers might generate diverse functions of SALMs, which could be the key for the regulation of neural wiring and function by SALMs.

Only SALM3 and SALM5 have synaptogenic activity in a *trans* manner among the SALM isoforms[24]. However, we found no marked structural difference in either the PTPδ-binding or dimerization interface between the presynapse-inducing SALM5 and the non-presynapse-inducing SALM2. On the other hand,

the positions of the bound PTPδ and SALM Ig relative to the SALM LRR dimers slightly differ between the PTPδ–SALM2 and PTPδ–SALM5 complexes, as shown in the superposition of these two complexes (Supplementary Fig. 8). This small difference in the orientation of the bound PTPδ might be relevant to the difference between the presynapse-inducing and non-presynapse-inducing SALMs, although further functional and structural analyses of the molecular mechanism for signal transduction from the extracellular domain to the cytosolic domain and downstream effectors such as liprin-α are needed.

Our structures revealed the 2:2 binding mode of PTPδ and SALMs. SALMs form a dimer, which bridges two PTPδ monomers. Although the interaction between PTPδ and SALMs is independent of the dimerization of SALMs, the dimerization of SALM5 is required for its synaptogenic activity. The dimerization of SALM5 may be necessary to reinforce the binding to the type-IIa RPTPs in neurons, because the affinity of PTPδ to SALM5 ($K_D = 14.4$ μM) is 26–96 times lower than those to other type-IIa RPTP-binding postsynaptic organizers (IL1RAPL1, 150 nM; IL-1RAcP, 510 nM; Slitrk2, 356 nM and TrkC, 551 nM)[15,16,30]. Moreover, the dimerization of SALM1–SALM3 potentially promotes the clustering of the proteins at the postsynaptic density through the interaction of their PDZ-binding motifs with PSD-95.

PTPδ localizes mostly in the axon terminal, whereas a recent study reported that the dendritic localization of PTPδ is promoted by the co-overexpression of PTPδ and IL1RAPL1[33]. In the same study, it was proposed that the *cis*-interaction between PTPδ and IL1RAPL1 mediates the recruitment of PTPδ to the postsynaptic membrane[33]. The present SPR analyses and synaptogenic assays using the site-directed PTPδ or SALM5 mutants showed that the PTPδ–SALM5 interaction and their dimer-of-dimer formation observed in the present PTPδ–SALM5 structure indeed occur under the physiological condition, likely in a *trans*-synaptic manner, although we cannot exclude the possibility that the

present PTPδ–SALM2 and PTPδ–SALM5 structures may also reflect the *cis* complex formed on the postsynaptic membrane.

The preference of SALMs for the splice variants of PTPδ was previously studied by cell-surface-binding assay, which suggested that the preference is different among SALMs[18,19]. On the other hand, our present structural and SPR analyses of SALM2 and SALM5 suggest that meA is not involved in the binding between the type-IIa RPTPs and SALMs, whereas the insertion of meB is preferred but dispensable for the binding. Further quantitative analysis is required to assess the preference for the splice variants of the type-IIa RPTPs, although preparations of other SALMs (SALM1, SALM3, and SALM4) for in vitro binding studies have been unsuccessful in our hands.

The Ig1 domain of the type-IIa RPTPs contains a proteoglycan-binding site. Heparan sulfate proteoglycans (HSPGs) and chondroitin sulfate proteoglycans promote and inhibit axon extension, respectively, through their interactions with the type-IIa RPTPs[29]. The crystal structures revealed that IL1RAPL1 and TrkC compete with proteoglycans for interaction with the type-IIa RPTPs[15,29,30]. Therefore, IL1RAPL1 and TrkC need to remove HSPGs from the type-IIa RPTPs to shift from the axon guidance state to the synapse formation state. In contrast, the SALM5-binding surface of PTPδ does not overlap with the proteoglycan-binding site, suggesting that SALM5 can bind to the type-IIa RPTPs regardless of its binding to HSPGs. This may be important for SALM5 to function in neurons, since the affinity of SALM5 to the type-IIa RPTPs is lower than those of other postsynaptic organizers.

To date, several postsynaptic organizer structures in complex with the type-IIa RPTPs have been reported[15–17,30] (Supplementary Fig. 1). Their bindings are mutually exclusive, possibly to sharpen the specificity to synapse targets[15]. This is also the case for SALM5. The notable feature of the PTPδ–SALM5 complex is its 2:2 stoichiometry, which is required for its synaptogenic activity. Molecular mechanisms of signal transduction by the 2:2 complex are important issues to be addressed in future studies on the type-IIa RPTPs and SALMs.

## Methods

**Protein preparation.** The cDNAs encoding mouse PTPδ Ig1–Ig3 (residues 28–325) and PTPδ Ig1–Fn1 (residues 28–418; A9B+, A9B– or A–B+) were cloned into the pEBMulti-Neo vector (Wako Pure Chemical Industries) with the N-terminal signal sequence derived from pHLsec vector[34]. The cDNAs encoding human SALM5 LRR–Ig (residues 18–378) and mouse SALM2 LRR–Ig (residues 32–390) are PCR amplified from a human cDNA library (Human Brain, whole QUICK-Clone cDNA, Clontech) and a mouse cDNA library (1st strand cDNA, Genostaff) and cloned into the pEBMulti-Neo vector with the N-terminal Igκ signal sequence and C-terminal His₆ tag. The cDNA encoding the entire ECD of human SALM5 lacking the signal sequence was cloned into pEB6-Igκ-Fc[4] to yield pEB6-Igκ-SALM5-Fc. The mutants of PTPδ Ig1–Fn1 (A9B+)-His₆, SALM5 LRR–Ig-His₆ and SALM5 ECD-Fc were produced by PCR-based site-directed mutagenesis. All proteins were transiently expressed in Expi293F cells (Thermo Fisher Scientific). The secreted proteins in the culture media were purified by Ni-NTA resin (Qiagen). The culture media containing the secreted proteins were loaded onto Ni-NTA resin equilibrated with 20 mM Tris-Cl (pH 7.4) containing 200 mM NaCl. The resin was then washed with 20 mM Tris-Cl (pH 7.4) containing 200 mM NaCl and 20 mM imidazole. The bound proteins were eluted with 20 mM Tris-Cl (pH 7.4) containing 200 mM NaCl and 250 mM imidazole. The purified PTPδ proteins were dialyzed against 20 mM Tris-Cl (pH 7.4) containing 150 mM NaCl. The SALM proteins were further purified by a Superdex200 10/300 size-exclusion column (GE Healthcare) with 20 mM Tris-Cl (pH 7.4) containing 250 mM NaCl.

**Crystallization.** For crystallization of apo-SALM5, SALM5 LRR–Ig was concentrated to 100 μM using an Amicon Ultra-4 filter (Millipore) and crystallized using the sitting drop vapor diffusion method at 20 °C by mixing 0.5 μL of the protein solution and 0.5 μL of the reservoir solution containing 10% PEG3350 and 0.2 M ammonium tartrate. For the crystallization of the PTPδ–SALM2 complex, SALM2 LRR–Ig was mixed with PTPδ Ig1–Ig3 at the final concentration of 57 μM each and co-crystallized using the sitting drop vapor diffusion method at 20 °C by mixing 0.5 μL of the complex solution and 0.5 μL of the reservoir solution containing 7% PEG20000 and 0.1 M MES (pH 6.0). For the crystallization of the

PTPδ–SALM5 complex, SALM5 LRR–Ig was mixed with PTPδ Ig1–Fn1 at the final concentration of 70 μM each and crystallized using the sitting drop vapor diffusion method at 20 °C by mixing 0.5 μL of the complex solution and 0.5 μL of the reservoir solution containing 7% PEG4000, 0.1 M NaCl, 0.1 M Li₂SO₄ and 0.1 M ADA (pH 6.5). The crystals of apo-SALM5 and the PTPδ–SALM2 and PTPδ–SALM5 complexes were cryoprotected by supplementation of the reservoir solutions with the final concentrations of 30% ethylene glycol, 35% PEG400 and 35% xylitol, respectively, and then flash-frozen in liquid N₂.

**Data collection and structure determination.** The diffraction data set of apo-SALM5 was collected at 100 K at beamline BL-1A of Photon Factory (Tsukuba, Japan). The diffraction data sets of the PTPδ–SALM2 and PTPδ–SALM5 complexes were collected at 100 K at beamline BL41XU of SPring-8 (Hyogo, Japan). The data sets were processed with HKL2000[35] and CCP4 program suite[36]. All structures were solved by the molecular replacement method using Molrep[37] or Phaser[38]. The LRR structure of NGL-3 (PDB 3ZYO)[39] was used as the search model for the structure determination of apo-SALM5. The apo-SALM5 structure and the previously determined PTPδ structure (PDB 4YFC and 4YFE)[15] were used as the search models for the structure determination of the PTPδ–SALM2 and PTPδ–SALM5 complexes. Model building and autocorrection/refinement were carried out using the programs Coot[40] and Phenix[41], respectively. Data collection and refinement statistics are shown in Table 1.

**SPR analysis.** SPR measurements were carried out using Biacore T200 (GE Healthcare) at 25 °C in 10 mM Hepes-Na (pH 7.5) containing 200 mM NaCl and 0.05% Tween20. SALM5 proteins were immobilized on a CM5 sensor chip by the amino-coupling method. PTPδ proteins were injected as analytes with adequate concentration series (0.1–260 μM) depending on the affinities. The sensorgrams are shown in Supplementary Figs. 2 and 4.

**SEC-MALS.** SALM proteins were concentrated to 0.5 g L⁻¹ and applied onto an ENrich SEC 650 (10 × 300 mm) column (Bio-Rad) in 20 mM Tris-HCl (pH 7.4) buffer containing 250 mM NaCl. The MALS data were collected by a DAWN HELEOS 8 + detector (Wyatt Technology) with an RF-20A UV detector (Shimadzu) and analyzed by the program ASTRA (Wyatt Technology).

**Synaptogenic assay.** Primary cerebral cortical cultures were prepared from mice at postnatal day 0. The cerebral cortices were treated with 1% trypsin and 0.1% DNase I in phosphate-buffered saline (PBS) for 5 min at room temperature. The cells were washed three times with Neurobasal A medium (Life Technologies) with 2% B-27 supplement (Life Technologies) and 5% fetal calf serum (Life Technologies) and dissociated by passing through a fire-polished Pasteur pipette in PBS containing 0.05% DNase I, 0.03% trypsin inhibitor and 2 mM MgCl₂. The cells were plated at a density of 2.5 × 10⁵ cells per cm² on glass cover slips (Matsunami glass) coated with poly-L-lysine (Sigma) and mouse laminin (Life Technologies). The cells were cultured in the Neurobasal A supplemented with 5% FCS, 2% B-27 supplement, 100 U mL⁻¹ penicillin, 100 μg mL⁻¹ streptomycin and 0.5 mM L-glutamine for 24 h, and then in the same medium without FCS. Expression vectors for mutated forms of human SALM5-Fc were generated by PCR-based mutagenesis using pEB6-Igκ-SALM5-Fc as a template. Fc and mutated forms of SALM5-Fc in FreeStyle 293-F cell culture media were bound to Protein A-conjugated magnetic particles (smooth surface, 4.0–4.5 μm diameter; Spherotech). Beads coupled with Fc or Fc-fusion proteins were added to cortical neurons on days in vitro 11. After 24 h, cultures were fixed for immunostaining with mouse anti-Bassoon antibody (Stressgen, 1:400) followed by Alexa555-conjugated donkey anti-mouse IgG (Invitrogen, 1:400) and FITC-conjugated donkey anti-human IgG (Jackson ImnmunoResearch, 1:400).

**Image acquisition and quantification.** For the quantitative measurements of Bassoon immunofluorescent signals, four or five optical images from two independent beads-neuron co-cultures were obtained using a confocal microscope (SP5II, Leica). Bassoon signal intensities on the beads were measured as the fluorescence mean density within a 7 μm diameter circle enclosing a coated bead. The fluorescent mean densities of the surrounding regions within a 14-μm diameter circle were then measured and subtracted as background signals. Statistical significance of difference was evaluated by one-way ANOVA followed by post hoc Tukey's test ($n = 28$–33 beads).

**Data availability.** The coordinates and structure factors of apo-SALM5 and the PTPδ–SALM2 and PTPδ–SALM5 complexes have been deposited in the Protein Data Bank under the accession codes 5XWS, 5XWU, and 5XWT, respectively. Primer sequences used in this study are listed in Supplementary Data 1. Other data are available from the corresponding authors upon reasonable request.

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

## Acknowledgements

We thank the beamline staff of the biological crystallography beamlines of Photon Factory (Tsukuba, Japan) and BL41XU of SPring-8 (Hyogo, Japan) for technical help during data collection. This work was supported by MEXT/JSPS KAKENHI (JP17K15072 to S.G.-I., JP16H04749 to A.Y., JP25290021 to T.U., JP25293057 to T.Y. and JP24247014 to S.F.), JST PRESTO (to T.Y.) and JST CREST (JPMJCR12M5 to T.U. and S.F.).

## Author contributions

S.G.-I., T.S., A.M., T.U., and T.Y. performed gene cloning, protein purification and crystallization. S.G.-I., A.Y., Y.S., and S.F. collected the diffraction data. S.G.-I., A.Y., and S.F. analyzed the collected data and determined the structures. S.G.-I. and T.S. performed the SPR measurements. A.I.-T. H.M. and T.Y. performed synaptogenic assay and T.S. analyzed the imaging data. S.G.-I., T.Y., and S.F. wrote the paper. T.Y. and S.F. designed and supervised the study.

## Additional information

**Competing interests:** The authors declare no competing financial interests.

