## [Peer Review File · Nature Communications]

Reviewers' comments:

Reviewer #1 (Remarks to the Author):

This manuscript reveals the molecular mechanisms underlying the synapse formation mediated by the trans-synaptic interaction between presynaptic PTPd and postsynaptic SALM2/5. The authors indentify a 2:2 complex formed by 2 PTPd and 2 SALM2/5 molecules, which is mediated by the Ig domains of PTPd and the LRR and Ig domains of SALM2/5. There are two independent sets of residues that mediate the PTPd and SALM2/5 interaction and the antiparallel dimerization between SALMs. Interestingly, in addition to the low-affinity PTPd-SALM5 interaction, the dimerization of SALM5 is important for SALM5-dependent presynaptic induction, suggesting that SALM5 dimerization may strengthen the PTPd-SALM5 interaction, and also multimerization of postsynaptic scaffolding proteins such as PSD-95. The PTPd-SALM5 interaction is unlikely to compete with previously known PTPd Ig interactions, further revealing the diverse modes of PTPd trans-synaptic interactions.

Given that the LAR family of PTPs is emerging as an interesting group of synaptic organizers, that it is important to understand how diverse trans-synaptic adhesions act in concert to orchestrate synapse development, and that SALM5 and PTPd have been associated with diverse psychiatric disorders, including autism, these new findings will be an important step forward in related fields. I have only the following minor points.

1. Although PTPd and SALM5 are likely to interact with each other in a trans-synaptic manner, a recent report by Motani et al (J Neurosci 2017) has suggested that PTPd may also be present at postsynaptic sites and that the identified PTPd-SALM5 interaction may also occur in a cis-manner on postsynaptic membranes. This should be discussed.
2. Figure 1b. It is not easy to appreciate how the whole structure of the 2:2 complex looks like from the images in Figure 1b. Perhaps the authors could show the whole 2:2 structure from the beginning (not just the half) and use different color schemes or some rotations to make the whole structure more easy to understand.
3. Figure 2. The blow-up images could be presented together with the overall structure so that the authors can easily understand where these enlarged structures are from.
4. Figure 3b,c. It would be nice if the authors could somehow indicate in figures where these enlarged structures are from Figure 3a (red, upper yellow, bottom yellow?).
5. Figure 3d. wt to WT (to be consistent with Figure 4).
6. Figure 4. How many experiments were performed, and how many images were used? Which statistical method was used?

Reviewer #2 (Remarks to the Author):

Manuscript: # 135371

Authors: Sakurako Goto-Ito, Atsushi Yamagata, Yusuke Sato, Takeshi Uemura, Tomoko Shiroshima, Asami Maeda, Ayako Imai-Tabata, Hisashi Mori, Tomoyuki Yoshida, and Shuya Fukai

Title of manuscript: Structural basis of trans-synaptic interactions between PTPδ and SALMs for inducing synapse formation

This manuscript presents the crystal structures of human SALM5 LRR-Ig alone, as well as human SALM5 and mouse SALM2 in complex with fragments of mouse PTP δ , namely Ig1-3+FN1 domains or PTP δ Ig1-3 both containing the mini-exon A and B inserts. SALMs and PTP δ are synaptic adhesion molecules that play a role in synapse development, and are associated with neuropsychiatric diseases. To substantiate the structural findings, the authors carry out biochemical experiments including structure-guided mutagenesis and heterologous synapse formation assays are carried out.

The key findings that this manuscript would like to present are the following:

- the structures of SALM5 (3.1 Ang) and its complex with PTP δ Ig1-3+FN1 (4.2 Ang), as well as the structure of SALM2 in complex with PTP δ Ig1-3 (3.2 Ang).
- while the Ig domain in SALM5 is likely flexible, in the complex with PTP δ Ig1-3+FN1, the Ig2 and Ig3 from PTP δ wrap around the SALM5 Ig domain fixing the latter's conformation. The same is seen in the complex between SALM2 and PTP δ Ig1-3.
- SALM5 exists as a dimer in solution, mediated by the LRR domains which pack in an antiparallel fashion
- dimerization of SALM5 is prerequisite for its functionality in inducing synaptic differentiation.
- SALM5 directly induces cis-dimerization of LAR-RPTPs into higher-order signaling assembly which mediate the clustering of bassoon (as a readout for presynaptic differentiation).

Significance: The mechanism of SALM2 and SALM5 function and their interaction with PTP δ has important implications for how synaptic adhesion molecules can promote the formation of synapses.

The structural work is beautiful. Structure-based mutants analyzed by well carried-out experiments involving SEC-MALS, SPR and cell-based assays suggest that 1) SALM5 is dimeric in solution, 2) the interaction between SALM5-PTP δ Ig1-Ig3+FN1 as seen in the crystal structure represents interactions in the physiological complex; key residues are also conserved, and 3) if you disrupt SALM5-PTP δ Ig1-Ig3+FN1 interaction or SALM5 dimerization, you prevent bassoon clustering (as a read-out for presynaptic differentiation).

Major comments:

1. Results section p. 7

"The PTP δ -interacting residues are conserved or functionally equivalent between SALM2 and SALM5 (Supplementary Table 1 and Supplementary Fig. 3). On the other hand, only SALM5 can induce presynaptic differentiation by binding to the type-IIa RPTPs19."

Please provide possible explanations why this might be in light of your structural studies.

2. Results section p. 10

"Furthermore, in the apo SALM5 crystal, a similar dimer-like interaction was also observed between two adjacent molecules related by crystallographic symmetry (Fig. 3a)."

Please provide a superposition between the SALM5 LRR domain alone compared to the SALM5 LRR domain in the complex so that the similarity can be assessed.

3. Table I

Please provide Ramachandran plot statistics (% residues in favorable region, allowed region and outliers) for each structure.

4. Please update Table I to reflect the statistics as provided in the PDB validation reports.

D_1300004291 3.08 Å. SALM5
D_1300004292 3.92 Å. protein A + protein B
D_1300004293 3.16 Å. protein A + protein B

5. Please provide brightfield images for Fig. 4a so that the reader can assess where the neurons are localized.

Minor comments:

1. Please have the manuscript thoroughly proofread. There are grammatical, typographical ("typos"), and stylistic errors.

2. Introduction section p. 3

"Dysfunctions of synaptic organizers potentially cause neurodevelopmental disorders such as autism spectrum disorders (ASD), intellectual disability or schizophrenia."

References are missing.

3. Introduction section p. 3

"Trans-synaptic interactions between pre- and postsynaptic organizers can induce synapse formation."

References are missing.

4. Introduction section p. 3

"Type-IIa receptor protein tyrosine phosphatases (RPTPs) and neuroligins are the two major presynaptic organizers^{1, 2}."

References are given for RPTPs only. Please add references for neuroligins.

5. Results section p. 7

"The Ig2 domain of PTP δ also interacts with the LRR domain of SALM2 or SALM5."

Please refer to Fig. 2 to aid the reader.

6. Results section p. 10

"The theoretical molar masses of the monomeric SALM2 and SALM5 LRR-Ig-His6 molecules are 40.2 and 41.2 kDa, respectively, whereas the molar masses of the SALM2 and SALM5 LRR-Ig-His6 molecules determined by SEC-MALS were 77.0 and 77.1 kDa, respectively, indicating that both SALM2 and SALM5 form homodimers in solution."

Please refer to Fig. 3d to aid the reader.

7. Please expand the legend for Figure 1c to better describe the contents.

8. Please indicate in the legend for Figure 3d what the dotted curves are.

Comments from Reviewer #1:

...Given that the LAR family of PTPs is emerging as an interesting group of synaptic organizers, that it is important to understand how diverse trans-synaptic adhesions act in concert to orchestrate synapse development, and that SALM5 and PTPd have been associated with diverse psychiatric disorders, including autism, these new findings will be an important step forward in related fields. I have only the following minor points.

1. Although PTPd and SALM5 are likely to interact with each other in a trans-synaptic manner, a recent report by Montani et al (J Neurosci 2017) has suggested that PTPd may also be present at postsynaptic sites and that the identified PTPd-SALM5 interaction may also occur in a cis-manner on postsynaptic membranes. This should be discussed.

The *cis*-interaction between PTP δ and SALM5 was discussed in the Discussion section (pg 14) as follows:

“PTP δ localizes mostly in the axon terminal, whereas a recent study reported that the dendritic localization of PTP δ is promoted by the co-overexpression of PTP δ and IL1RAPL1 (Montani *et al.*, *J. Neurosci.*, 2017). In the same study, it was proposed that the *cis*-interaction between PTP δ and IL1RAPL1 mediates the recruitment of PTP δ to the postsynaptic membrane (Montani *et al.*, *J. Neurosci.*, 2017). The present SPR analyses and synaptogenic assays using the site-directed PTP δ or SALM5 mutants showed that the PTP δ -SALM5 interaction and their dimer-of-dimer formation observed in the present PTP δ -SALM5 structure indeed occur under the physiological condition, likely in a *trans*-synaptic manner, although we cannot exclude the possibility that the present PTP δ -SALM2 and PTP δ -SALM5 structures may also reflect the *cis* complex formed on the postsynaptic membrane.”

Montani *et al.*, *J. Neurosci.*, 2017 was cited in the discussion accordingly.

2. Figure 1b. It is not easy to appreciate how the whole structure of the 2:2 complex looks like from the images in Figure 1b. Perhaps the authors could show the whole 2:2 structure from the beginning (not just the half) and use different color schemes or some rotations to make the whole structure more easy to understand.

In Fig. 1b, two views of the whole 2:2 complex from different angles were presented with different color schemes for each 1:1 PTP δ -SALM complex accordingly.

3. Figure 2. The blow-up images could be presented together with the overall structure so that the authors can easily understand where these enlarged structures are from.

In addition to the blow-up images, the overall structure of PTP δ -SALM5 complex was also presented as Fig. 2a. The regions corresponding to the blow-up images (Fig. 2b, c and d) were enclosed by rectangles in Fig. 2a.

4. Figure 3b,c. It would be nice if the authors could somehow indicate in figures where these enlarged structures are from Figure 3a (red, upper yellow, bottom yellow?).

In Fig. 3a, the central and peripheral dimer interfaces were enclosed by rectangles with labels ('b' and 'c') to indicate that the central and peripheral interfaces correspond to Fig. 3b and Fig. 3c, respectively.

5. Figure 3d. wt to WT (to be consistent with Figure 4).

'wt' was changed to 'WT' accordingly.

6. Figure 4. How many experiments were performed, and how many images were used? Which statistical method was used?

Four or five optical images from two independent co-cultures (28-33 beads for each mutant or wild type) were used for the quantitative measurements of Bassoon immunofluorescent signals. Statistical significance was evaluated by one-way ANOVA followed by *post hoc* Tukey's test. This information was described in the Methods section. The statistical method used in this study and the numbers of beads used for the statistical analyses were also described in the legend for Fig. 4.

Comments from Reviewer #2:

... *Significance: The mechanism of SALM2 and SALM5 function and their interaction with PTP δ has important implications for how synaptic adhesion molecules can promote the formation of synapses.*

The structural work is beautiful. Structure-based mutants analyzed by well carried-out experiments involving SEC-MALS, SPR and cell-based assays suggest that 1) SALM5 is dimeric in solution, 2) the interaction between SALM5-PTP δ Ig1-Ig3+FN1 as seen in the crystal structure represents interactions in the physiological complex; key residues are also conserved, and 3) if you disrupt SALM5-PTP δ Ig1-Ig3+FN1 interaction or SALM5 dimerization, you prevent bassoon clustering (as a read-out for presynaptic differentiation).

Major points:

1. Results section p. 7

“The PTP δ -interacting residues are conserved or functionally equivalent between SALM2 and SALM5 (Supplementary Table 1 and Supplementary Fig. 3). On the other hand, only SALM5 can induce presynaptic differentiation by binding to the type-IIa RPTPs.”

Please provide possible explanations why this might be in light of your structural studies.

This point was discussed in the Discussion section (pg 13-14) as follows:

“Only SALM3 and SALM5 have synaptogenic activity in a *trans* manner among the SALM isoforms. However, we found no marked structural difference in either the PTP δ -binding or dimerization interface between the presynapse-inducing SALM5 and the non-presynapse-inducing SALM2. On the other hand, the positions of the bound PTP δ and SALM Ig relative to the SALM LRR dimers slightly differ between the PTP δ -SALM2 and PTP δ -SALM5 complexes, as shown in the superposition of these two complexes (Supplementary Fig. 8). This small difference in the orientation of the bound PTP δ might be relevant to the difference between the presynapse-inducing and non-presynapse-inducing SALMs, although further functional and structural analyses of the molecular mechanism for signal transduction from the extracellular domain to the cytosolic domain and downstream effectors such as liprin- α are needed.”

Our trials to change SALM5 to a non-presynapse-inducing SALM without affecting its dimerization and binding to PTP δ by site-directed mutation have been unsuccessful so far.

2. Results section p. 10

“Furthermore, in the apo SALM5 crystal, a similar dimer-like interaction was also observed between two adjacent molecules related by crystallographic symmetry (Fig. 3a).”

Please provide a superposition between the SALM5 LRR domain alone compared to the SALM5 LRR domain in the complex so that the similarity can be assessed.

The superposition between the SALM5 LRR domain alone and that in complex with PTP δ was presented in Supplementary fig. 5a.

3. Table I

Please provide Ramachandran plot statistics (% residues in favorable region, allowed region and outliers) for each structure.

Ramachandran plot statistics were added in Table 1.

4. Please update Table I to reflect the statistics as provided in the PDB validation reports.

D_1300004291 3.08 Å. SALM5
D_1300004292 3.92 Å. protein A + protein B
D_1300004293 3.16 Å. protein A + protein B

Table 1 was updated to reflect the statistics as provided in the latest PDB validation reports. The PDB file of the PTP δ -SALM5 complex (PDB 5XWT) shows that the length of the *c*-axis is 210.946 Å. We therefore show that the length of the *c*-axis of the PTP δ -SALM5 crystal is 210.9 Å in Table 1, although the PDB validation report shows that it is 210.95 Å. The completeness values were calculated by the program HKL2000 (Scalepack).

5. Please provide brightfield images for Fig. 4a so that the reader can assess where the neurons are localized.

The bright field images were added in Fig. 4a.

Minor points:

1. Please have the manuscript thoroughly proofread. There are grammatical, typographical (“typos”), and stylistic errors.

We have carefully read and checked the manuscript for English usage. In addition, the manuscript was further checked by a commercial English editing service for scientific papers.

2. Introduction section p. 3

“Dysfunctions of synaptic organizers potentially cause neurodevelopmental disorders such as autism spectrum disorders (ASD), intellectual disability or schizophrenia.”

References are missing.

The following four papers were cited in this sentence (pg 3):

Betancur *et al.*, *Trends Neurosci* **32**, 402–412 (2009).

Chen *et al.*, *Front Cell Neurosci* **8**, 276 (2014).

Takahashi and Craig, *Trends Neurosci* **36**, 522–534 (2013).

Um and Ko, *Trends Cell Biol* **23**, 465–475 (2013).

3. Introduction section p. 3

“Trans-synaptic interactions between pre- and postsynaptic organizers can induce synapse formation.”

References are missing.

The following two papers were cited in this sentence (pg 3):

Siddiqui and Craig, *Curr Opin Neurobiol* **21**, 132–143 (2011).
McMahon and Díaz, *Curr Opin Neurobiol* **21**, 221–227 (2011).

4. Introduction section p. 3

“Type-IIa receptor protein tyrosine phosphatases (RPTPs) and neuroligins are the two major presynaptic organizers1, 2.”

References are given for RPTPs only. Please add references for neuroligins.

The following three papers were additionally cited in this sentence (pg 3):

Siddiqui and Craig, *Curr Opin Neurobiol* **21**, 132–143 (2011).

McMahon and Díaz, *Curr Opin Neurobiol* **21**, 221–227 (2011).

Craig and Kang, *Curr Opin Neurobiol* **17**, 43–52 (2007)

5. Results section p. 7

“The Ig2 domain of PTPδ also interacts with the LRR domain of SALM2 or SALM5.”

Please refer to Fig. 2 to aid the reader.

We referred to Fig. 2 in this sentence (pg 7).

6. Results section p. 10

“The theoretical molar masses of the monomeric SALM2 and SALM5 LRR–Ig–His6 molecules are 40.2 and 41.2 kDa, respectively, whereas the molar masses of the SALM2 and SALM5 LRR–Ig–His6 molecules determined by SEC-MALS were 77.0 and 77.1 kDa, respectively, indicating that both SALM2 and SALM5 form homodimers in solution.”

Please refer to Fig. 3d to aid the reader.

We referred to Fig. 3d in this sentence (pg 10).

7. Please expand the legend for Figure 1c to better describe the contents.

The descriptions of the color-coding of each domain and the disulfide bonds were added in the legend for Fig. 1c as follows:

*“N-cap, LRR, C-cap and Ig are colored in pink, cyan, purple and grey, respectively.
The disulfide bonds are shown as yellow sticks.”*

8. Please indicate in the legend for Figure 3d what the dotted curves are.

The description for the dotted lines was added in the legend for Fig. 3d as follows:

“The plotted dots that look like dotted lines correspond to the calculated molar masses of each protein.”